# ECG Authentication Based on Non-Linear Normalization under Various Physiological Conditions

**DOI:** 10.3390/s21216966

**Published:** 2021-10-20

**Authors:** Ho Bin Hwang, Hyeokchan Kwon, Byungho Chung, Jongshill Lee, In Young Kim

**Affiliations:** 1Department of Biomedical Engineering, Hanyang University, Seoul 04763, Korea; hobin0215@hanyang.ac.kr; 2Information Security Research Division, Electronics and Telecommunications Research Institute (ETRI), Daejeon 34129, Korea; hckwon@etri.re.kr (H.K.); cbh@etri.re.kr (B.C.)

**Keywords:** biometrics, authentication, various physiological conditions, ECG, non-linear, normalization

## Abstract

The development and use of wearable devices require high levels of security and have sparked interest in biometric authentication research. Among the available approaches, electrocardiogram (ECG) technology is attracting attention because of its strengths in spoofing. However, morphological changes of ECG, which are affected by physical and psychological factors, can make authentication difficult. In this paper, we propose authentication using non-linear normalization of ECG beats that is robust to changes in ECG waveforms according to heart rate fluctuations in various daily activities. We performed a non-linear normalization method through the analysis of ECG alongside heart rate, evaluating similarities and authenticating the performance of our new method compared to existing methods. Compared with beats before normalization, the average similarity of the proposed method increased 23.7% in the resting state and 43% in the non-resting state. After learning in the resting state, authentication performance reached 99.05% accuracy for the resting state and 88.14% for the non-resting state. The proposed method can be applicable to an ECG-based authentication system under various physiological conditions.

## 1. Introduction

Security technology is essential for the function of common convenience facilities such as banks, airports, corporate buildings, and websites [1]. Authentication means accessing to use services that have security. The authentication methods we commonly use include passwords, tokens, and ID cards. These traditional methods may inconvenience users by requiring them to remember their passcodes. These methods are also vulnerable to spoofing caused by loss, theft, and squinting [2]. Many biometric authentication methods have been proposed to address this shortcoming.

Biometrics includes physiological and behavioral traits used to distinguish people statistically and is simpler and more secure than popular existing authentication methods [3]. Physiological traits include external information about the body, such as fingerprints, iris, face, and veins, and internal information, such as the electrocardiogram (ECG), electromyography (EMG), and brain wave (EEG) patterns. Behavioral traits include habit-based information such as voice, gait, and signature [4]. Besides that, the combination of biometrics has been studied in order to have the most robust information [5]. Although many biometrics are widely used in devices and services, they are vulnerable to spoofing attempts. For example, fingerprint and iris recognition can be disabled with artificial fingerprints made of rubber materials or special films and infrared photos [6,7]. Facial and vein recognition can also be misrecognized by the photos of enrolled subjects and replicas printed with special materials [8,9]. However, ECG is robust against spoofing attempts and can replace traditional biometrics due to the following characteristics: it is invisible externally, it can only be measured by physical contact, and it is unique to each individual [10]. In addition, ECG-based biometrics can be used in various devices with the development of devices capable of real-time measurement and the increase in research on protecting authentication information in devices [11,12].

The ECG is an electrical signal acquired through an electrode attached to the skin and consists of a P wave, QRS complex, and T wave [13]. There are three reasons why ECG patterns vary from person to person [14]. The first is a physiological factor that reflects heart size, mass, conductivity, and activity, all of which differ from person to person. The second is a geometrical factor reflecting the difference between the position and vector of the heart. Finally, DNA affects the detailed shape and composition of the heart individually. However, since ECG is an electrical signal, it is greatly affected by the measurement environment and changes in heart rate. The effect of heart rate negatively affects authentication performance. To eliminate this effect, many researchers are interested in normalization.

In the authentication system, it is preferred to acquire a signal in a short time. The number of ECG beats within a short time is limited, and one beat is critical. Therefore, correcting one beat to eliminate heart rate effects is an important task, but most of studies have not considered compensating for ECG changes with various heart rates [15,16,17,18,19]. Among the studies considering some heart rate, many previous investigations resampled the entire ECG waveform by substituting the RR-interval into a linear equation. Additionally, it is usually used to remove outlier beats by analyzing the correlation between a representative waveform and other waveforms [20,21,22,23,24]. Arteaga-Falconi et al. [21] derived a normalization technique that divides the duration of the P wave, QRS complex, and T wave of the ECG by the duration of the RR-interval. They obtained an 81.82% true acceptance rate (TAR) and 1.41% false acceptance rate (FAR) from 10 subjects with ECG sensors embedded in smartphones. These approaches considered only noise removal and morphological characteristics and did not compare results from ECGs taken under various conditions.

Other prior research normalized only a part of the ECG to improve biometric performance. Fatemian et al. [25] reconstructed only the T wave section to 120 *msec* based on the standard duration of an approximate T wave in the resting state and achieved a maximum recognition rate of 95% for 21 subjects. Choi et al. [26] proposed a method to match the P wave, QRS complex, and T wave, which are morphological characteristics of the ECG after stepper exercise, with the pre-exercise cycle to improve recognition performance. They obtained 96.4% accuracy and an improved mean similarity of 23.5% for 100 subjects by comparing ECG waveforms before and after exercise. Nobunaga et al. [27] designed an optimized band-pass filter and normalized the ECG measured after exercise. They used ECG data measured post-exercise in 10 subjects and obtained 99.7% accuracy.

Most of the mentioned studies did not consider various situations and did not focus on validating the normalization. Few studies have considered some situations, but only one of the physiological conditions and have missed heart rate information. This point makes it difficult to determine whether the proposed system is applicable in real life. On the other hand, our study acquired data that were confirmed heart rate changes in various states and quantitatively evaluated effectiveness and authentication performance according to normalization by comparing existing methods.

In real life, ECG-based authentication would usually be performed under various physiological and mental conditions. Even in such conditions, the variability of ECG within individuals should be minimized. Therefore, this paper aims to improve authentication performance under various conditions by minimizing variability in individuals. For this purpose, the main contributions of this work are as follows. First, we developed a non-linear normalization method. Our method improved the accuracy and robustness of an authentication system using ECG that changes morphologically according to various heart rates. Second, we constructed a database of measurements in various conditions that can be experienced in real life: rest, exercise, listening to music, and watching video clips. Through this database, the effectiveness of the proposed method was tested by comparing the similarity and authentication performance between the proposed method and the existing methods. Finally, we discussed the feasibility that the ECG-based authentication system can be applied in real life.

The contents after Section 1 are as follows. Section 2 presents our time normalization method and user recognition system, and Section 3 presents normalization performance and authentication results using ECG under various conditions. Section 4 discusses the results and limitations of this study. Section 5 summarizes the research and presents future research.

## 2. ECG Authentication Algorithm Based on Non-Linear Normalization

Biometric systems identify or authenticate the person based on one or more characteristics. Three blocks, a signal collection block that collects signals with sensors, a storage module that stores the registered subject’s data, and a biometric algorithm block, constitute a traditional biometric system.

We designed the framework of the proposed work based on the traditional frame. Figure 1 shows the workflow of the framework for ECG authentication system proposed in this study. The measured signal is transmitted to the authentication module through the data processing module. The enrollment process generates a trained user model with the extracted features, and the authentication process is performed by inputting the extracted features into the enrollment model. The framework is divided into a data processing module that minimizes individual data variability before authentication and a user authentication module that performs authentication with processed data. The data processing module consists of the following steps: a pre-processing module for denoising the ECG; ECG fiducial detection and segmentation module; Non-linear normalization module to minimize individual variability. The user authentication module consists of the following steps: extracting features from normalized ECG sequences; A model design module for user authentication with the extracted features and an authentication module for user evaluation. All stages of the proposed framework are explained in detail in the following.

### 2.1. Database

The database consisted of lead I ECG measurements obtained using a Biopac MP150 (Biopac System Inc., Goleta, CA, USA), AcqKnowledge 4.2, and an ECG100C ECG amplifier with the sampling rate set at 500Hz and the measurement gain to 1000. Subjects comprised of 15 healthy adults without heart disease, 13 males and 2 females (mean age 27.29 years, SD 2.58 years). Subjects completed ECG measurement experiments over periods of 2–11 months. The IRB of Hanyang University approved this study, and all subjects supplied informed consent before the experiment (HYI-16-030-1).

Table 1 describes the six measurement environments, such as resting, exercise, listening to music (calm and excited), and watching video clips (relaxed and scared). We measured ECG records for 20 s while the subject sat in a chair during each randomly assigned situation or immediately after exercise. The measurement time referred to the ECG-ID database that was applied in various studies [28]. We obtained 1969 records from 15 subjects, including a wide range of beats per minute (51–132 bpm) and six statuses that reflect normal daily life activities.

### 2.2. Pre-Processing

The acquisition of ECG through electrodes is hindered by noises, such as baseline drift, power line interference, motion artifacts, and electromyography. These noises compromise the original signal. If authentication is performed using noisy ECG data, performance will be degraded, so it is essential to remove noises.

We used a notch filter to suppress 60 Hz power line interference alongside a 3rd-order Butterworth high pass filter with a cutoff frequency of 0.5 Hz to filter Baseline wandering. Motion artifacts and electromyography were removed based on the Daubechies-6 wavelet function at decomposition level 4, the soft thresholding method, and a wavelet denoising algorithm applying the *Bayes-Shrinkage* rule [29,30].

### 2.3. Fiducial Points Detection and Segmentation

As shown in Figure 2, ECG has five types of fiducial points (P, Q, R, S, and T) in the normal state and three types each of intervals (PR interval, QT interval, ST interval) and segments (PR segment, ST segment, TP segment). In ECG, three main components, the P wave, QRS complex, T wave, are repeated periodically. Atrial depolarization causes the p wave, ventricular depolarization causes the QRS complex, and ventricular repolarization causes the T wave.

It is convenient to identify fiducial points to analyze ECG signals using the wavelet function at different scales. Many previous studies have employed an algorithm described previously by Martinez et al. [31]. This algorithm decomposes ECG at multiple scales and pinpoints P wave onset (*P_on_*), QRS complex onset (*QRS_on_*), R peak (*R_peak_*), QRS complex offset (*QRS_on_*), and T wave offset (*T_off_*) by finding modulus maxima, minima, and zero-crossing points. They decomposed the signal by selecting a quadratic spline as the mother wavelet and setting the scale to five, to capture the fiducial points. The results we obtained through this algorithm are shown in Figure 3a.

Subsequently, we defined one beat as a fixed interval from *P_on_* to *P_on_*, as shown in Figure 3b. Unlike cutting and using a fixed-length window, this method detected one beat and included the complete waveform between *P_on_* and *P_on_*.

### 2.4. Non-Linear Normalization Based on Heart Rate Analysis

Heart rate changes depending on physical activity or psychological state can cause heartbeat cycle changes in the electrocardiogram. Simoons et al. [32] found that ECG changes with non-linear characteristics through interval analysis of ECG alongside heart rate. Craig et al. [33] observed that the control and response of the cardiac cortex are asymmetrical because the asymmetrical autonomic nerves of the heart display asymmetric distributions. In other words, these distributions cause asymmetry in the ventricular and atrial functions and control. Therefore, we proposed non-linear normalization through segment-by-segment analysis according to heart rate (HR).

In this study, we analyzed the ECG-ID database of Physionet [28]. The ECG-ID database contains 310 ECG recordings of 44 males and 46 females between 13 to 75 years old. Each of the 90 subjects had at least two or more 20 sec records [34]. The ECG of the database was segmented into one defined beat and divided into four sections: PR interval (Pon-QRSon), QRS complex (QRSon-QRSoff), ST interval (QRSoff-Toff), and TP segment (Toff-Pon). We performed a regression analysis of heart rate for each section. Figure 4 shows the regression results and scatter plots for each section.

Consequently, the heart rate had little effect on the QRS complex (Coefficient of determination ($r^{2}$) = 0), and the remaining sections showed high correlations with heart rate in the order of TP segment ($r^{2}=0.9$), ST interval ($r^{2}=0.7$), and PR interval ($r^{2}=0.4$). To examine whether there are further non-linear changes with heart rate in the ST interval and TP segment, which have higher correlations with heart rate, we further segmented these intervals by piecewise non-linear regression and analyzed changes in duration according to heart rate [35]. This method performed normalization by dividing the number of segments in the ST interval and TP segment into 1, 2, and 3. Table 2 shows the analysis results with the average root mean square error (RMSE) value, and it shows similar results even if the number of segments is increased compared to the case of dividing into one.

As shown in Table 2, the normalization of *ST* interval shows an error of 0.0338 when the number of segments piecewise is 1 or 2 and an error of 0.0339 when the number of segments piecewise is 3. The normalization in the *TP* segment has an error of 0.0249 when the number of segments piecewise is 1, 2, and 3. Therefore, the piecewise average RMSE values confirmed that further segmentation of the ST interval and TP segment did not significantly affect normalization. Based on the regression equation analyzed above, normalization was performed by resampling each section as much as the standard duration at 70 bpm, the resting heart rate. To compensate for ECG amplitude, we applied z-score normalization [36].

Figure 5 shows the flowchart of the algorithm to correct each segment of ECG with the analysis result. As shown in Figure 5, the proposed normalization method is modified with time when HR is 70 bpm for a total of four sections. After correction, we stored the normalized ECG beat in a variable, and this process is repeated as many as the number of R peaks in the ECG data.

This flowchart leads to Algorithm 1. In the algorithm, *PR interval, QRS complex, ST interval,* and *TP segments* were replaced with PR, QS, ST, and TP. The input is *N_R_*, the number of R peaks, and *X* consists of ECG beats. The algorithm iterates *N_R_* times and copies four sections to the output array *Seq’*. In order to perform normalization for each beat, we extracted four sections from the *i*-th beat and checked if the time was at 70 bpm for each section. If it needs to be modified, it is interpolated with time at 70 bpm. When all sections are corrected, normalized beats are returned.
**Algorithm 1:** The proposed normalization algorithm
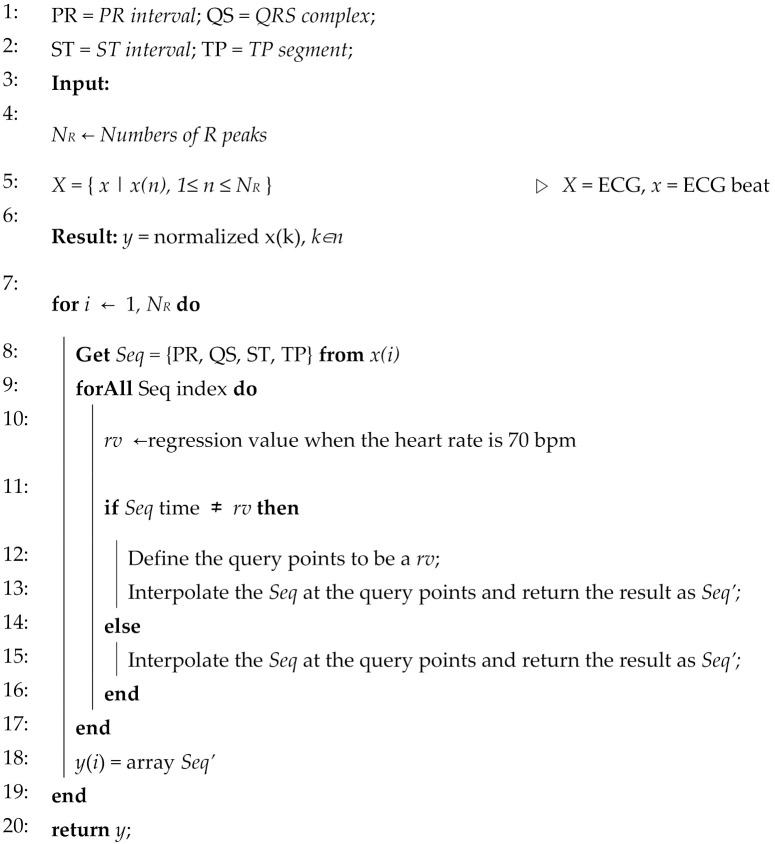


Figure 6 shows the results of applying the conventional linear normalization method and the proposed non-linear normalization using 132 ECG data samples with the wide bpm range (57–120 bpm) from subject 11, who was measured over a period of 8 months in this study.

### 2.5. Similarity Measurement of Normalization

Even if normalization is performed, whether the normalized beat is similar to the ECG beat in the resting state is unknown. To evaluate this, we generated the representative beat in the resting state and extracted the similarity to beats of resting and non-resting states that were not used to generate this beat. This process was applied to the existing and proposed methods to compare differences in the similarity between methods.

For resting status, we randomly selected data for each subject to generate the representative beat, then grouped the remaining and non-resting status data as evaluation data. We gradually increased the number of generating data files from five to 20. To avoid deviating significantly from the general morphology of an ECG beat, the beats in this data were aligned, and the outlier beats were removed as proposed by Tan et al. [37]. The beats of the generating data overlapped with the median values of R peaks extracted from itself. Figure 7 shows the process of detecting outliers and generating representative beat. We compared the Pearson correlation coefficients beats of the generating data ($ECG_{all}$) with the temporary beats ($ECG_{tmp}$) constructed as the average of *ECG_all_*. These coefficients were produced as input to the conditions (Equation (1)), and the beats satisfying the condition were considered outliers. The outlier condition is as follows:(1)$$\rho \left(ECG_{all},\;ECG_{tmp}\right)<\mu_{\rho }-0.5\times \sigma_{\rho },$$
where ρ is the correlation coefficient, $\mu_{\rho }$ is the mean of ρ, and $\sigma_{\rho }$ is the standard deviation of ρ.

Outlier beats were excluded when constructing the representative beats. To compare how many beats were properly modified, we extracted the proportion of outliers removed for each method. Then, the representative beat was constructed by calculating the average of the remaining beats. The similarity before and after normalization was evaluated through cross-correlation with representative beats and beats of evaluation data.

### 2.6. Authentication System

#### 2.6.1. Feature Extraction

Feature extraction aims to enhance individual uniqueness from divided beats while minimizing the effects of noise and user variability. ECG studies generally grouped feature extraction methods into fiducial and non-fiducial approaches. Fiducial approaches are a combination and information of fiducial points such as differences in duration, amplitude, or slopes. Non-fiducial approaches use a specific model or transformation from the original signal to another domain. However, there are problems with non-fiducial approaches. These approaches increase the computational load when authentication uses transformation coefficients and depend on the signals, even for an ECG waveform containing much noise [38].

In this study, we performed authentication with fiducial features that express the heart’s unique anatomical structure and have robust noise properties. We extracted a total of 29 features through the combination of previously extracted P wave, QRS complex, and T wave [38]. Features 1 to 9 are amplitude features between fiducial points, 10 to 18 are duration on the time axis, 19 to 28 are slope and distance, and 29 is area. Figure 8 shows features extracted from the ECG beat with relative amplitude, temporal duration, slope, distance, and area through fiducial points and summarizes the extracted features.

#### 2.6.2. Authentication

Although authentication and identification seem similar, they have different meanings, and accordingly, the classifier models used for each are different. Identification uses multiple classes to classify, whereas authentication uses binary classes for each subject. Therefore, we performed authentication by creating a single model for each class [39].

We used the support vector machine (SVM) based on Gaussian kernel (SVM-G), SVM based on quadratic kernel (SVM-Q), k-nearest neighbor (kNN), and decision tree (DT) as an authentication model. When designing the model, the parameters’ values for each model were set as follows: SVM-G and SVM-Q were used to optimize the gamma value for each individual, and the C value was fixed at 1. The K parameter in the kNN was set to 1. The DT parameters were min_samples_leaf = 3 and min_samples_split = 10. These models were applied with 10-fold cross-validation and were implemented in a one-vs-rest approach that allows us to classify multi-class labels as a binary classification problem for each class [39].

We designed two different authentication experiments and tested our proposed method. The first experiment selected an optimal model. The second experiment, based on the optimal model, evaluated the proposed method in various measurement environments. Figure 9 shows data partitioning for two experiments. Both experiments used the same train set, and in the first experiment, we randomly selected a data set in the resting state for each subject and split it according to an 80:20 ratio into the train set and the test set for model selection. The classifier with the best authentication performance was selected as the optimal model among the four classifiers using this train set and test set.

In the second experiment, we prepared a test set under several different statuses for evaluations of authentication performance. For each status, the test set consists of the registrant’s data and the other’s data, which is the same size as the registrant, using randomness control. We evaluated the authentication performance between the existing and proposed methods through this test set and the selected optimal model.

All the tests were run on standard lab equipment (Intel(R) Core(TM) i7-7700 CPU, 3.60 GHz, 32 GB of RAM, One RTX 2080Ti 11 GB) or in MATLAB’s Statistics and Machine Learning Toolbox.

## 3. Results

In this section, we described the quantitative analysis results of the proposed non-linear normalization and the performance of the proposed authentication method.

### 3.1. Evaluation of Normalization

#### 3.1.1. Outlier Analysis

Figure 10 shows the results of comparing ratios of beats recognized as outliers depending on each normalization method to compare and evaluate the performance of existing methods. We used independent t-tests when the Shapiro-Wilk normality test confirmed normal distributions. Otherwise, the Mann–Whitney test was used. All statistical analyses were performed using IBM SPSS Statistics 24.0 for Windows (IBM, Armonk, NY, USA).

Our proposed method showed significant differences according to methods recommended by Arteaga-Falconi et al. [21] (*t*-test, *** *p* < 0.001), Fatermian et al. [25] (*t*-test, *** *p* < 0.001), and Nobunaga et al. [27] (*t*-test, *** *p* < 0.001) and Choi et al. [26] (Mann–Whitney test, ** *p* < 0.01). The existing methods discarded more beats than the presented method when generating the representative beat. Our proposed method did not detect as many outlier beats as existing methods and better maintained the number of beats than other methods. This result indicates that our proposed method corrected more beats to resemble the resting state waveform than existing methods.

#### 3.1.2. Quantitative Similarity Analysis

Table 3 shows correlation coefficient values of ECG beats in the resting states. To confirm the performance of our proposed method, we compared the proposed method to existing methods. When generating the representative beat, differences may occur depending on the number of data files. Therefore, we compared results with the different numbers of data files to generate the representative beat. In Table 3, when the number of data files generating a representative beat is 5 and 20, the difference in correlation coefficient differs by 0.0098 on average. This result means that even if the data file increases, the correlation coefficient does not change significantly. Therefore, using the representative beat constructed with five files, we compared correlation coefficients between our proposed and existing normalizations methods.

When comparing the similarity of our method presented with existing methods, our method showed superior performance (ranging from 4.6% to 17.7%) compared to existing methods (Table 3). In addition, similarity increased by 23.7% for ECG beats without normalization. These results should be interpreted with caution because the method used by Choi et al. did not detect the clear difference in performance from the proposed method, compared to other methods, perhaps because beats were compared in the resting state.

Table 4 shows correlation coefficient values between the resting state and the non-resting state. The similarity was compared depending on the number of files in the same way as the resting state. There were no significant differences in the correlation coefficient according to how many files were included.

Compared to the existing methods, our method presented showed better performance (ranging from 2.6% to 22%) (Table 4). In addition, similarity increased 42.5% using the proposed method for beats of the ECG without normalization. This result is a more considerable performance difference than that observed between resting states. In particular, our method showed a difference of −0.3% compared to the Choi et al. method in the resting state but a better result by 3.1% in the non-resting state.

### 3.2. Authentication Performance

To evaluate the proposed method, we selected the optimal model by comparing the performance of authentication models. We used accuracy (ACC), False Acceptance Rate (FAR), False Rejection Rate (FRR), and an Equal Error Rate (EER) as performance indicators [40]. FAR evaluates results masquerading as a class other than the intended class, whereas FRR evaluates results erroneously rejected as members of a particular class. EER is an index that evaluates the reliability of the model with the location of the receiver operating characteristic (ROC) curve when FAR and FRR are the same.

We extracted the results by changing only the authentication model under the same conditions to select the most appropriate model. Figure 11 and Table 5 show the performance parameters of ACC, FAR, FRR, and EER for each authentication model in the resting state. The results were obtained using the train set in the resting state for each individual and the test set for model selection.

Figure 11 and Table 5 illustrate that the proposed method demonstrates better performance in SVM-G than other authentication models. SVM-G showed better average differences in accuracy, 1.1% to kNN, 0.5% to SVM-Q, and 3.3% to DT. This model led to better FAR and EER to all normalization methods, ranging from a minimum of 1.6% to a maximum of 7%. Although SVM-G showed 0.12% lower performance on average in FRR than the kNN model, it had better performance for indicators other than FRR. In addition, SVM-G is more efficient in terms of security due to a lower FAR. Our proposed method achieved 99.05% ACC, 0.85% FAR, 1.04% FRR, and 0.77% ERR in SVM-G and brings good performance for all four performance indicators when compared to other methods. In addition, the proposed method showed similar or higher performance than existing methods in other models.

We analyzed the percent change of heart rate to classify non-resting values according to heart rate. Figure 12 shows the percent change of heart rate for each status based on the heart rate in the resting state. Listening to the calm music and watching the scary videos resulted in the lowest heart rate change of 5–6%, while listening to the exciting music and watching the relaxed videos resulted in the second-highest heart rate change of 7–8%, and exercise resulted in the highest heart rate change of 48%.

Figure 13 and Table 6 illustrate the performance parameters of the SVM-G model in the non-resting state. The results were obtained using the train set in the resting state for each individual and the test set to compare authentication performances according to state. When all conditions were used as the test set, our method obtained 88.14% ACC, 2.6% FAR, 21% FRR, and 9.83% EER, resulting in superior results to existing methods. When authentication is divided into three stages, authentication performance deteriorates as the percent change of heart rate increases. For calm music and scary video, the proposed method showed better accuracy than existing methods at 93.55%. In the relaxed video and exciting music, the proposed method earned good results of accuracy 90.87%.

When the test set used only exercise status, the accuracy of the proposed method was higher than other methods, but the authentication performance was lower than other statuses. This result means that the high heart rate variability may significantly impair the signal, making heart rate corrections ineffective. Therefore, when the change in heart rate is 10% or more compared to the current heart rate, authentication performance is severely degraded, and authentication may not proceed normally.

## 4. Discussion

### 4.1. Non-Linear Normalization Based on Analysis with Heart Rate

The accuracy of ECG-based authentication systems was previously unconfirmed at different heart rates, but this study can serve as the precedent for using authentication systems in real life. We proposed the novel normalization method and used ECG data obtained in various states to evaluate our method’s performance and analyzed correlations with ECG beats extracted during different physiological states. This process yielded that our method improved the similarity of ECG beats before and after normalization compared to existing methods and showed good authentication performance.

Previous studies of ECG-based authentication systems have proposed various ideas to improve authentication systems’ accuracy. However, only a few studies employed corrections for heart rate variability in ECG. For example, Li et al. [20] implemented an algorithm to normalize the entire length of an ECG beat at a fixed length to test authentication models. However, they did not include conditions that affect physiological heart rate changes in ECG. Arteaga-Falconi et al. [21] designed and verified an authentication system by linearly performing time normalization. However, this approach has the disadvantage of not considering the non-linear characteristics of ECG. We addressed such shortcomings in this study. We performed non-linear normalization for each section, considering ECG nonlinearity. In addition, we performed the specific evaluation of our method compared to existing methods.

Some previous studies have attempted to correct ECG morphological information by performing time-normalization of only T wave, P wave, and T wave parts using partial normalization [25,26]. However, these studies conducted normalization based on commonly used physiological parameters rather than empirical analysis and did not perform experimental verification of heart rate changes.

The significant advantage of our study is that it presents scientific evidence from statistical analysis of the wide range of heart rate measurements. This evidence was obtained by modifying each section based on the section-by-section analysis of ECG according to heart rate. In addition, we performed experimental validation on the proposed method with the database that provided information on the percent change of heart rate. The following discussion describes that this database extracted at different states is valuable for evaluating its authentication performance.

### 4.2. Authentication Performance with Heart Rate Changes

As for the results of SVM-G, the proposed method in the resting state shows better FAR, FRR, and accuracy than existing methods (Table 5). However, authentication accuracy decreases as the percent change of heart rate increases (Table 6). The performance deteriorates the most in the exercise state, during which the heart rate increases significantly, and the change in FRR value becomes the greatest compared to other indicators. Nevertheless, the FAR value was 2.61% for the complete training process, indicating that the indicator is still highly resistant to the approach of others. The increase in the FRR value indicates the point where the individual waveform does not match the resting state even after time normalization.

We performed further analysis and confirmed the results through feature selection to find the reason. For the feature selection algorithm, we used the Relief-F algorithm. The Relief-F input consisted of resting and exercising state features for each individual. Since the number of resting state data and exercise state data is different, the resting state data were randomly selected to the number of exercise state data. We earned the feature weights that summed were summed for all subjects per feature after z-score normalization.

Figure 14 shows the summed feature weights per feature and that the amplitude features are in the Top 8: the amplitude of QT, PS, RS, PT, ST, RQ, and PR. A high ranking meaning is a dominant feature when distinguishing the resting state from the exercising state. In other words, the performance degradation in the exercising state comes from the difference in amplitude. In addition, the temporal domain features corrected by the proposed method are ranked low. A low ranking means that the feature is stable in any state. The feature selection result implies that specific normalization of amplitude is required along with non-linear normalization in the case of exercising. Several previous studies support these results. Previously published studies explored whether the amplitude values of the P wave, T wave, and QRS complex change during rapid heart rate changes [32,41,42]. Langley et al. [43] described that when amplitude increased due to the significant change in heart rate, it was restored to the relaxed heart rate after a recovery period of 300 sec. There are z-score and min-max normalization methods for amplitude normalization, but it is challenging to correct ECG waveforms due to the high degree of heart rate change, so it is not suitable for normalization of amplitude that changes according to heart rate [19].

We evaluated authentication performance in various situations, with results constituting interesting potential for the emergence of the new normalization method that considers amplitude rather than only the time axis during high heart rates. These insights could be applied to developing a new authentication system that can accommodate excessive increases in heart rate. In addition, our results can guide future research for application to real life.

### 4.3. ECG Based Authentication System Including Various Real-Life Conditions

Our study experimented with fewer subjects compared to other studies. However, we constructed the database with ECG data measured by performing various physiological statuses similar to real life. We observed variability in authentication performance over a wide range (53–132 bpm) of heart rates. Observation of authentication performance according to heart rate changes may provide a cornerstone for future study of ECG authentication.

Research using a large database with many subjects can increase the thoroughness of performance evaluation, as the diversity of subjects is guaranteed [19]. However, the inclusion of many subjects does not ensure diversity of health conditions. This shortcoming makes it difficult to evaluate whether a method tested in resting can be applied in real life. For example, Kim et al. [44] designed and tested a system using single-channel ECG signals measured from 11 males for 10 min over six days. However, they only performed experimental validation using data measured in the resting state and did not consider signals measured in real-life conditions. Choi et al. [26] studied 100 subjects engaging in stepper exercise and verified differences in authentication performance associated with an activity. Since this study does not specify heart rate after exercise, it is difficult to confirm heart rate changes due to exercise.

One of the significant advantages of our study is that we evaluated the practical use of the authentication system implemented as a database that was designed and measured for several situations. This evaluation method can minimize problems that may occur in the authentication system when implemented in real life. We considered more diverse situations compared with the studies mentioned above. This merit allows us to evaluate the feasibility of implementing an authentication system in a clinical setting. However, it was difficult to determine precisely when authentication performance is reduced due to sharp changes in heart rate, so this should be noted.

## 5. Conclusions

In this paper, we proposed normalization along a time axis to improve ECG authentication and evaluated a data set in various states. We artificially changed the heart rate by listening to music, watching a video, or exercising to authenticate users in various situations. Then, one ECG cycle composed of a P wave, QRS complex, and T wave, which are morphological characteristics of the ECG, was analyzed. Our non-linear normalization method was applied by analyzing the ECG cycle before the heart rate change.

We evaluated user authentication performance under various conditions in a sample of 15 subjects by using the proposed method. In the resting state, our proposed method improved the similarity before and after the heart rate change by 8% on average compared to existing methods, and our approach achieved 99.05% ACC, 0.77% ERR authentication performance. These indicators of performance are better than other methods in SVM-G. In the other state, our proposed method improved similarity 11% on average compared to existing methods and achieved 88.14% ACC, 9.83% ERR authentication performance across all states.

Future research needs to evaluate authentication performance in various conditions using a large sample. Performance should be maintained when used in any state. There is a need for an amplitude and time normalization method to avoid detrimental authentication performance even with significant changes in heart rate. In that respect, our results suggest directions for future studies. Deep learning can be a powerful alternative, but difficulty in the practical application has to be taken into account by the inconvenience of the increased computational load. Therefore, the normalization step seems to be more important and should be addressed more in future studies.

## Figures and Tables

**Figure 1 sensors-21-06966-f001:**
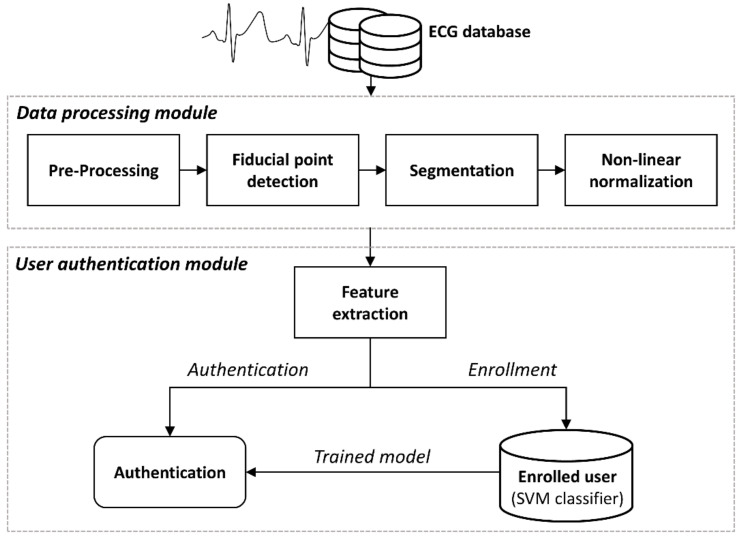
Workflow of the proposed framework for ECG based user authentication.

**Figure 2 sensors-21-06966-f002:**
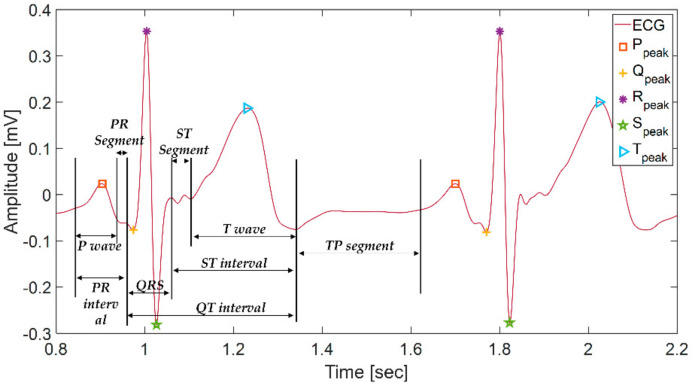
ECG waveform and morphological features.

**Figure 3 sensors-21-06966-f003:**
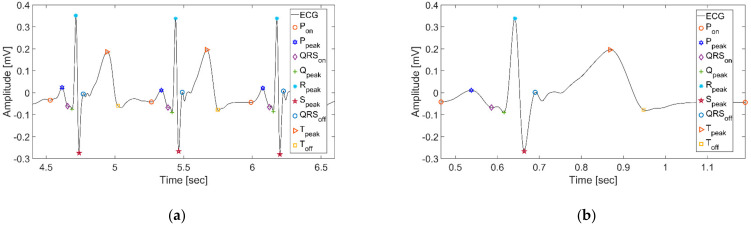
Fiducial point detection and segmentation based on ECG: (**a**) Detection of fiducial points in ECG based on Wavelet transform; (**b**) Segmentation results based on *P_on_* points.

**Figure 4 sensors-21-06966-f004:**
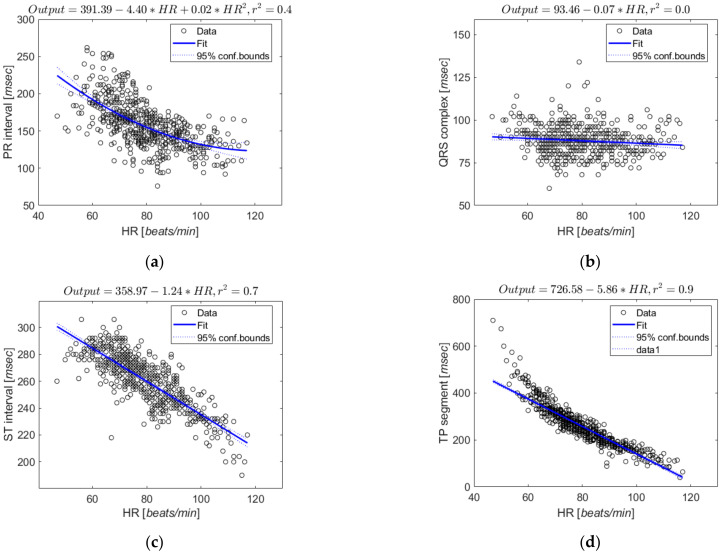
Regression plots for ECG segments and intervals at various heart rates: (**a**) PR interval; (**b**) QRS complex; (**c**) ST interval; (**d**) TP segment.

**Figure 5 sensors-21-06966-f005:**
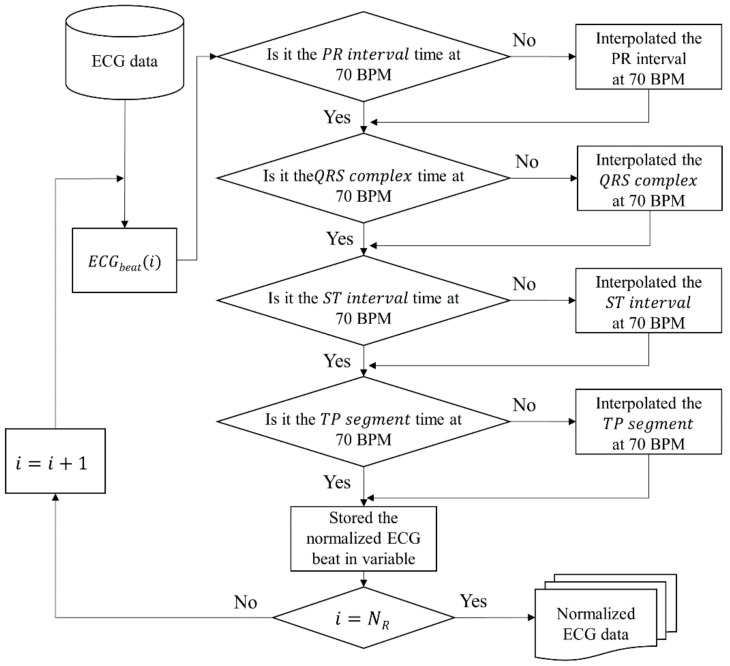
Flowchart of the proposed normalization algorithm.

**Figure 6 sensors-21-06966-f006:**
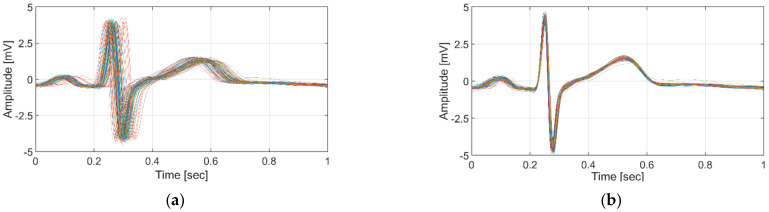
Normalization result for subject 11 with a 57–120 bpm heart rate: (**a**) with linear normalization; (**b**) with the proposed method.

**Figure 7 sensors-21-06966-f007:**
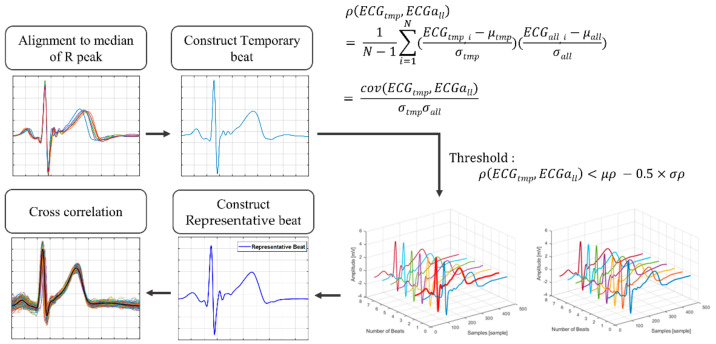
The process of generating the representative beat after removing outlier beats.

**Figure 8 sensors-21-06966-f008:**
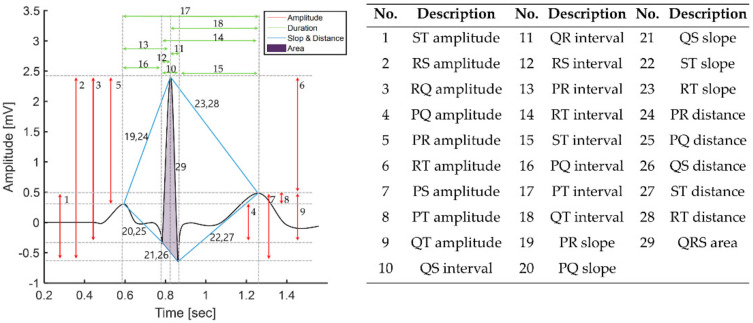
Fiducial features on the single ECG beat and the table that summarizes features.

**Figure 9 sensors-21-06966-f009:**
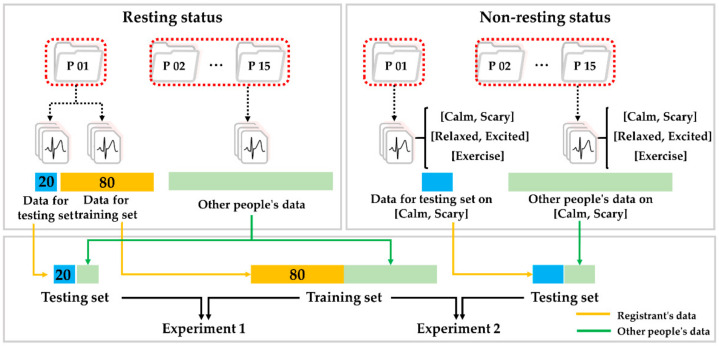
Schematic diagram of data partitioning method for two experiments; Example of test set configuration for experiment 1 of subject 1 (**left**); Example of test set configuration using only calm and scary status for experiment 2 of subject 1 (**right**).

**Figure 10 sensors-21-06966-f010:**
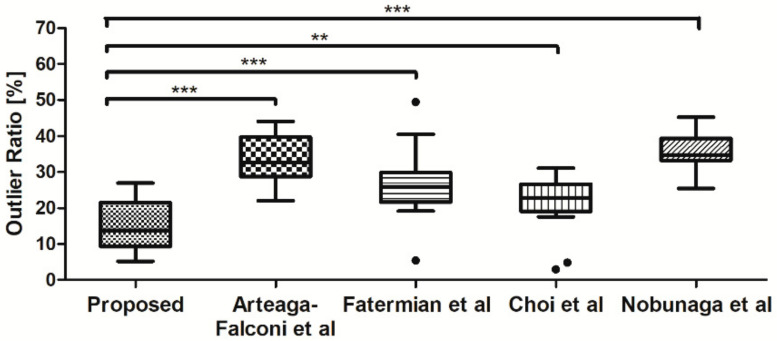
Ratios of beats recognized as outliers according to each method; Proposed method (mean: 14.84, SD: 6.65), Arteaga-Falconi et al. (mean: 33.30, SD: 6.84), Fatermian et al. (mean: 26.46, SD: 9.37), Choi et al. (median: 22.84, IQR: 1.00), and Nobunaga et al. (mean: 35.52, SD: 4.63); ***: *p* < 0.001, **: *p* < 0.01, ●: Outlier data.

**Figure 11 sensors-21-06966-f011:**
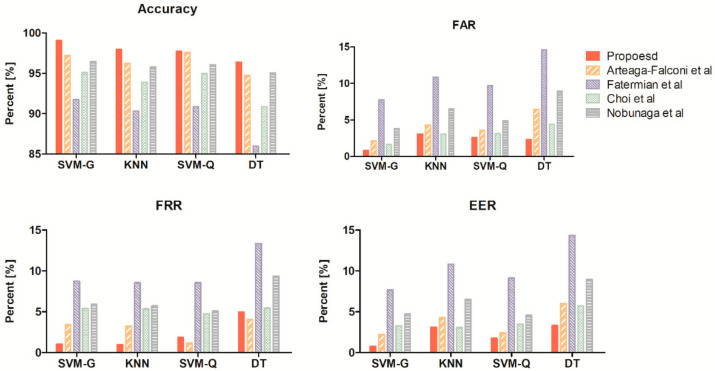
Evaluation indicator of authentication for each machine learning model.

**Figure 12 sensors-21-06966-f012:**
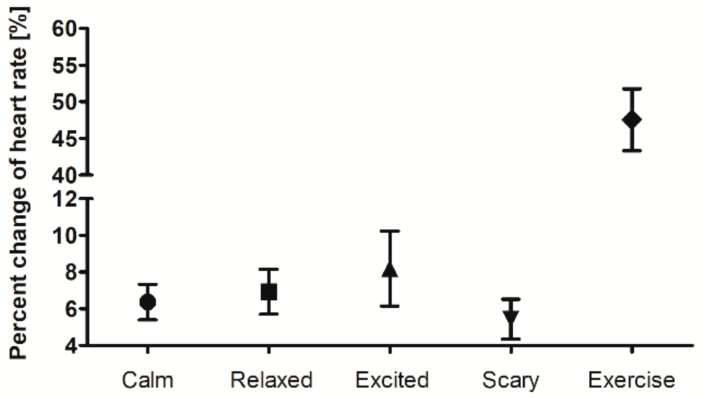
Percent change in heart rate for each status compared to the resting state.

**Figure 13 sensors-21-06966-f013:**
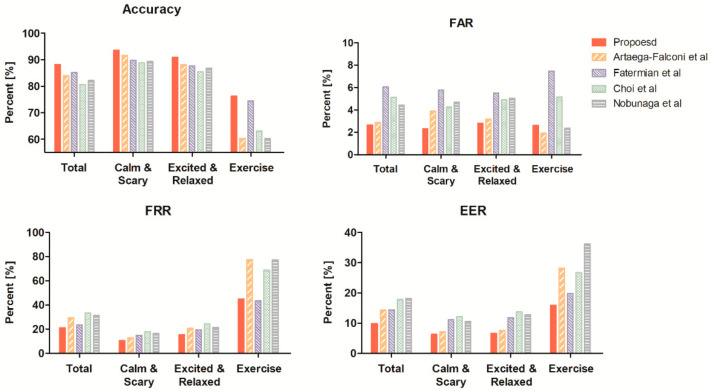
Evaluation indicator of authentication for each status.

**Figure 14 sensors-21-06966-f014:**
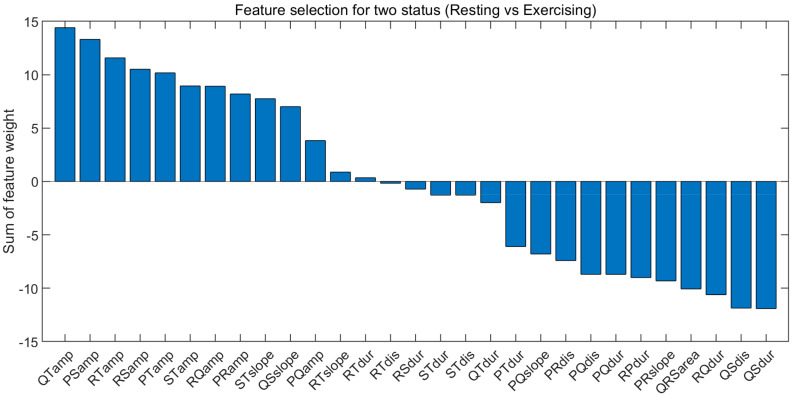
Sum of feature weight to all subjects.

**Table 1 sensors-21-06966-t001:** Detailed descriptions for each status.

Status	Description
Resting	Five-minute rest in sitting position
Listening to music	Calm: Listening to classical music wearing a headset Excited: Listening to rock music wearing a headset
Watching video	Relaxed: Watching the animated video Scared: Watching horror video
Exercising	Five times round trip at maximum speed, climbing stairs as high as the second floor

**Table 2 sensors-21-06966-t002:** Average RMSE value depending on piecewise regression divided by the number of each segment.

	$\mathrm{ST}\mathrm{Interval}\;(QRS_{off}-T_{off})$	$\mathrm{TP}\mathrm{Segment}\;(T_{off}-P_{off})$
One-segment piecewise regression	0.0338	0.0249
Two-segment piecewise regression	0.0338	0.0249
Three-segment piecewise regression	0.0339	0.0249

**Table 3 sensors-21-06966-t003:** Comparisons of performance with the correlation coefficient between proposed method and existing methods in the resting state.

State	Resting State					
Files	Raw	Arteaga- Falconi et al. [21]	Fatermian et al. [25]	Choi et al. [26]	Nobunaga et al. [27]	Proposed
**5**	**0.7868**	**0.8747**	**0.9300**	**0.9765**	**0.8267**	**0.9729**
10	0.7931	0.8793	0.9270	0.9787	0.8401	0.9756
15	0.7895	0.8695	0.9268	0.9788	0.8540	0.9759
20	0.7934	0.8767	0.9289	0.9795	0.8711	0.9772
**Relative error for similarity**	23.7%	11.2%	4.6%	−0.3%	17.7%	

**Table 4 sensors-21-06966-t004:** Comparisons of performance between proposed method and existing methods in the non-resting state.

State	Non-Resting State					
Files	Raw	Arteaga- Falconi et al. [21]	Fatermian et al. [25]	Choi et al. [26]	Nobunaga et al. [27]	Proposed
**5**	**0.6611**	**0.7717**	**0.9178**	**0.9136**	**0.7858**	**0.9418**
10	0.6760	0.7948	0.9179	0.9223	0.8051	0.9464
15	0.6723	0.7769	0.9169	0.9180	0.8059	0.9460
20	0.6695	0.7879	0.9185	0.9210	0.8292	0.9465
**Relative error for similarity**	42.5%	22%	2.6%	3.1%	19.9%	

**Table 5 sensors-21-06966-t005:** The average values of accuracy, FAR, FRR, and EER for each machine learning model.

	Method	SVM-G	kNN	SVM-Q	DT
Accuracy	**Proposed**	**99.05**	**97.97**	**97.76**	**96.38**
	Arteaga-Falconi et al. [21]	97.20	96.23	97.62	94.72
	Fatermian et al. [25]	91.77	90.31	90.88	86.02
	Choi et al. [26]	96.47	95.77	96.07	95.07
	Nobunaga et al. [27]	95.1	93.88	95.00	90.84
FAR	**Proposed**	**0.85**	**3.09**	**2.61**	**2.28**
	Arteaga-Falconi et al. [21]	2.17	4.28	3.59	6.48
	Fatermian et al. [25]	7.74	10.83	9.71	14.62
	Choi et al. [26]	1.67	3.08	3.10	4.39
	Nobunaga et al. [27]	3.86	6.52	4.91	8.95
FRR	**Proposed**	**1.04**	**0.98**	**1.88**	**4.96**
	Arteaga-Falconi et al. [21]	3.43	3.26	1.18	4.08
	Fatermian et al. [25]	8.73	8.54	8.53	13.33
	Choi et al. [26]	5.39	5.38	4.77	5.47
	Nobunaga et al. [27]	5.93	5.72	5.09	9.36
EER	**Proposed**	**0.77**	**3.09**	**1.74**	**3.32**
	Arteaga-Falconi et al. [21]	2.23	4.28	2.41	6.02
	Fatermian et al. [25]	7.67	10.83	9.12	14.33
	Choi et al. [26]	3.27	3.08	3.47	5.69
	Nobunaga et al. [27]	4.72	6.52	4.56	8.95

FAR (false acceptance rate); FRR (false rejection rate); EER (equal error rate).

**Table 6 sensors-21-06966-t006:** The average values of accuracy, FAR, FRR, and EER, for each status.

	Method	Total	Calm Music and Scary Video	Excited Music and Relaxed Video	Exercise
Accuracy	**Proposed**	**88.14**	**93.55**	**90.87**	**76.15**
	Arteaga-Falconi et al. [21]	83.88	91.64	88.10	60.28
	Fatermian et al. [25]	85.23	89.71	87.63	74.46
	Choi et al. [26]	80.66	88.82	85.40	63.14
	Nobunaga et al. [27]	82.12	89.38	86.82	60.22
FAR	**Proposed**	**2.66**	**2.33**	**2.79**	**2.61**
	Arteaga-Falconi et al. [21]	2.88	3.90	3.17	1.92
	Fatermian et al. [25]	6.07	5.77	5.50	7.48
	Choi et al. [26]	5.12	4.27	4.90	5.15
	Nobunaga et al. [27]	4.45	4.69	5.04	2.37
FRR	**Proposed**	**21.06**	**10.57**	**15.47**	**45.09**
	Arteaga-Falconi et al. [21]	29.36	12.81	20.63	77.53
	Fatermian et al. [25]	23.46	14.80	19.25	43.61
	Choi et al. [26]	33.55	18.09	24.31	68.58
	Nobunaga et al. [27]	31.30	16.56	21.32	77.19
EER	**Proposed**	**9.83**	**6.23**	**6.62**	**15.86**
	Arteaga-Falconi et al. [21]	14.29	7.24	7.65	28.13
	Fatermian et al. [25]	14.40	11.08	11.81	19.82
	Choi et al. [26]	17.82	12.1	13.66	26.70
	Nobunaga et al. [27]	18.10	10.57	12.76	36.15

FAR (false acceptance rate); FRR (false rejection rate); EER (equal error rate).

## Data Availability

Data sharing is not applicable to this article.

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
