# Peer review of "ECG Authentication Based on Non-Linear Normalization under Various Physiological Conditions"

_sensors, 2021, doi:10.3390/s21216966_

Round 1
Reviewer 1 Report
In this paper, the authors have proposed authentication using non-linear normalization of ECG beats that is robust to changes in ECG waveforms according to heart rate fluctuations in various daily activities. The paper is written well. My comments are given as follows.
- Please highlight the contributions of your work in the introduction section.
- The advantages and the disadvantages of the work should be written.
- PLease add citations in Table 5 and Table 6.
Reviewer 2 Report
Research Topic is interesting and emerging, but authors are highly recommended to improve the paper according to the following comments
Minor Comments
- Flow and overall organization of the paper is fine
- Research topic/idea is interesting
Major Comments
- What is the motivation of the proposed work?
- Introduction needs to explain the main contributions of the work clearer.
- The novelty of this paper is not clear. The difference between present work and previous Works should be highlighted.
- Authors must explain in detail the introduction section.
- Authors must develop the framework/architecture of the proposed methods
- There is need of flowchart and pseudocode of the proposed techniques
- Proposed methods should be compared with the state-of-the-art existing techniques
- Research gaps, objectives of the proposed work should be clearly justified.
- To improve the Related Work and Introduction sections authors are highly recommended to consider these high quality research works <‘ An Energy-Efficient Algorithm for Wearable Electrocardiogram Signal Processing in Ubiquitous Healthcare Applications”, MDPI Sensors Vol.8, No.3, pp.923, 2018'>, <'A Lightweight Portable Intrusion Detection Communication System for Auditing Applications’, International Journal of Communication Systems,Wiley, Article ID: DAC4327, Vol. 33, no 7, Jan 2020 '>
- English must be revised throughout the manuscript.
- Limitations and Highlights of the proposed methods must be addressed properly
- Experimental results are not convincing, so authors must give more results to justify their proposal.
Finally, paper needs major improvements
Reviewer 3 Report
It is a well-presented manuscript, which addresses an important and interesting approach of the research problem. I suggest some changes that could be taken into consideration.
In line 31, possibly it would be better to change into: remember their possess codes.
Possibly, you could include that the combination of Facial recognition, galvanic skin response and cardiac pulse has been studied in order to have a most robust information, in other areas of analysis such as sensory food analysis.
Álvarez-Pato VM, Sánchez CN, Domínguez-Soberanes J, Méndoza-Pérez DE, Velázquez R. A Multisensor Data Fusion Approach for Predicting Consumer Acceptance of Food Products. Foods. 2020; 9(6):774. https://doi.org/10.3390/foods9060774
Revise line 59, it is not well cited. Arteaga-Falconi et al. [21] (¿?)
In line 75, would these results be conclusive or preliminary if they only take into account 10 subjects? This is worth of consideration, because your study takes into account only 15 subjects, from which 2 are women? However the analysis of 90 subjects’ data within Physionet is well performed. And at the end you establish a method for implementation in the future. This issue is answered in lines 386, 418-420. However, possibly in the introduction (line 75 you could explain a bit more this issue)
Figure 1 is well presented, however the information of Training of Data model is very small and can´t be easily read.
In line 110, please revise the word scary (scared?)
In Table 1, possibly the correct words are: watching? Climbing stairs?
Is the time that the subject was imposed to these stimuli important in your measurements? Why did you choose 20 sec ?
In line 151 I would eliminate the following words… and it.
In line 206, do you refer to seconds?
The results are well presented, and easily understood.
In line 306, the idea is not clear, possibly: even if the numer of data files increases the ...... And possibly you could show evidence of this.
Round 2
Reviewer 2 Report
Authors have significantly revised the paper, so I recommend acceptance in its present form